# Errors in Broadband Permittivity Determination Due to Liquid Surface Distortions in Semi-Open Test Cell [†]

**Michał Kalisiak *** and **Wojciech Wiatr**

Institute of Electronic Systems, Warsaw University of Technology, 00-665 Warsaw, Poland; wiatr@ise.pw.edu.pl

* Correspondence: M.Kalisiak@elka.pw.edu.pl; Tel.: +48-22-234-7663

† This paper is an extended version of our paper published in Proceedings of the 23rd International Microwave and Radar Conference (MIKON), IEEE, Warsaw, Poland, 5–7 October 2020 by M.K. and W.W.: Analysis of Meniscus Impact on Broadband Liquid Permittivity Determination Based on Electromagnetic Simulations.

**Abstract:** We study how surface distortions of liquid samples due to a meniscus and a tilt of a semi-open coaxial test cell affect errors in a broadband permittivity determination. The study is based on the scattering parameters, obtained using the electromagnetic simulations of samples with flat and distorted surfaces in a broad frequency range up to 18 GHz. The parameters are processed with the classic Nicolson–Ross–Weir (NRW) method and our new meniscus removal technique. We analyze the errors for several samples of different properties, such as distilled water and propan-2-ol. The results show that the meniscus removal technique is more robust and provides smaller errors in the permittivity determination compared to the classic NRW method. The effect of the cell tilt, to our best knowledge, has not been considered in the literature yet.

**Keywords:** complex permittivity; microwave measurements; electromagnetic simulations; scattering parameters; errors; meniscus; tilt; skew

## 1. Introduction

The accurate value of liquid complex permittivity $\varepsilon_r = \varepsilon_r' - j\varepsilon_r''$ is desired in many fields of science and technology [1] such as medicine, biology, chemistry, agriculture, radio communication, remote sensing [2–4], etc. The permittivity, according to [5,6], represents the interaction of a material with an electric field. Its real part $\varepsilon_r'$ is related to the energy stored, while the imaginary part is tied in with the dielectric loss and conductivity of a material due to an external field. Depending on the frequency of interest, the permittivity is measured using various techniques [1,5,7,8], as e.g., capacitance method [9], resonant cavities [10], free space measurements [11], time-domain spectroscopy [12], transmission/reflection (T/R) and reflection-only methods [13,14].

At microwave frequencies, the most popular and reliable are T/R and resonant methods. Although the latter ones allow very precise permittivity determination, as the resonant frequency and the quality factor of the cavity are highly sensitive to the permittivity, the characterization is limited only to a narrow bandwidth around one or just a few frequencies. In contrast, the T/R methods allow broadband permittivity characterization by measuring liquid samples in test cells with vector network analyzers (VNAs). Permittivity affects the propagation constant ($\gamma$) and the impedance of the wave guiding cell and thus can be determined from the complex scattering parameters measured. However, the accuracy of such measurements heavily depends on the test cell design and a method applied to the calculation.

Among different test cells for two-port broadband measurements, vertically oriented semi-open cells feature great versatility [15–17]. Clogged at the bottom with a dielectric plug, they allow adjusting the volume of a liquid to provide the optimal measurement conditions regarding e.g., signal attenuation. However, the top of the liquid sample is distorted, due

to the surface tension, forming a meniscus [18]. Such a surface deviation from the flat and transversal may cause errors in the permittivity determination [15,16,19–21]. A meniscus formed around a brass rod exemplifying the boundary between water and the cell center conductor is presented in Figure 1.

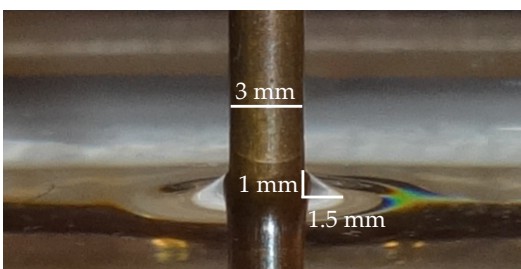

**Figure 1.** Photo of a meniscus formed around a brass rod exemplifying the boundary between water and the cell center conductor.

To get rid of the meniscus effect, the authors of [22–24] applied a cell closed with two plugs that fix the volume and shape of the liquid sample. However, such a solution causes more problems with modeling the measurement and constrains its application to a predefined volume of liquid.

To overpass problems with the meniscus in semi-open cells, which are more universal, we have proposed a new method [25] that can remove systematic errors caused by the meniscus in the broadband permittivity measurements. Its high performance has been confirmed experimentally [25]. However, due to the finite uncertainty of the scattering parameters measurement with a calibrated VNA, it was uneasy to get deeper into errors that occur when the conventional techniques as [15,16,26] are applied to this goal.

In this paper, we study the errors caused by the meniscus in the permittivity and sample column height determination. To this end, we employ the scattering parameters obtained with electromagnetic (EM) simulations of the liquid samples that do not suffer from the residual errors of VNA calibration. We also analyze errors due to a not perfectly vertical position of the cell. The effect of the cell tilt, to our best knowledge, has not been considered in the literature yet. To check the impact of the liquid surface distortions on the errors, we investigate two different algorithms: our meniscus removal (MR) method and the classic Nicolson–Ross–Weir (NRW) technique [26,27].

## 2. Measurement Procedures

In this section, mathematical algorithms used in the data processing are presented. We start from modeling the simulated measurement states. In Section 2.1 we present a classical approach for permittivity calculation with one state of liquid using the NRW algorithm [26,27] and Somlo's method [15] for the sample height determination. In Section 2.2 we summarize our meniscus removal (MR) method, presented in [25], that uses two volumes of liquid.

In measurements using semi-open coaxial cells, a liquid sample is kept in place by a dielectric plug—Figure 2b. S-parameters of such the structure are measured with a calibrated VNA, with reference planes at the connectors of the fixture. The whole accessory to measure the S-parameters with a VNA will be called here "the fixture", and the part of the fixture above the plug as "the cell". Effects of the plug and bottom part of the fixture can be eliminated mathematically [16] with the additional measurement of the empty cell, shown in Figure 2a.

Such sample measurements are typically described with the transfer matrices (**T**) which are related to the **S** matrix as follows:

$$\mathbf{T} = \begin{bmatrix} T_{11} & T_{12} \\ T_{21} & T_{22} \end{bmatrix} = \frac{1}{S_{21}} \begin{bmatrix} -\det \mathbf{S} & S_{11} \\ -S_{22} & 1 \end{bmatrix}, \qquad \mathbf{S} = \frac{1}{T_{22}} \begin{bmatrix} T_{12} & \det \mathbf{T} \\ 1 & -T_{21} \end{bmatrix}. \tag{1}$$

The measurement of the empty fixture $\mathbf{T}_{f0}$ can be expressed as the transmission through the airline section and the bottom part of the fixture (see Figure 2a):

$$\mathbf{T}_{f0} = \mathbf{T}_{a0}\mathbf{T}_b , \tag{2}$$

while the state with the liquid sample (Figure 2b):

$$\mathbf{T}_{f1} = \mathbf{T}_{a1}\mathbf{T}_{s1}\mathbf{T}_b , \tag{3}$$

where

$$\mathbf{T}_{ak} = \begin{bmatrix} e^{-\gamma_a l_{ak}} & 0 \\ 0 & e^{\gamma_a l_{ak}} \end{bmatrix} \tag{4}$$

describes the transmission through the airline section of the length $l_{ak}$ and the known propagation constant $\gamma_a$, and

$$\mathbf{T}_{s1} = \mathbf{T}_t \begin{bmatrix} e^{-\gamma_s l_{s1}} & 0 \\ 0 & e^{\gamma_s l_{s1}} \end{bmatrix} \mathbf{T}_t^{-1} \tag{5}$$

is the transmission through the sample of height $l_{s1}$ referenced to the airline impedance. $\mathbf{T}_t$ represents the transition from the air to the sample, $\gamma_a$ and $\gamma_s$ are the propagation constants for the air and the sample, respectively. To eliminate the bottom part of the fixture, we use the empty cell, getting the representation of the test cell itself—the part above the plug, representing only the sample and the airline section above it:

$$\mathbf{T}_{c1} = \mathbf{T}_{a1}\mathbf{T}_{s1} = \mathbf{T}_{f1}\mathbf{T}_{f0}^{-1}\mathbf{T}_{a0} . \tag{6}$$

The length of the empty cell $l_{a0}$, needed in (6), is usually known from the mechanical or electrical measurements [28].

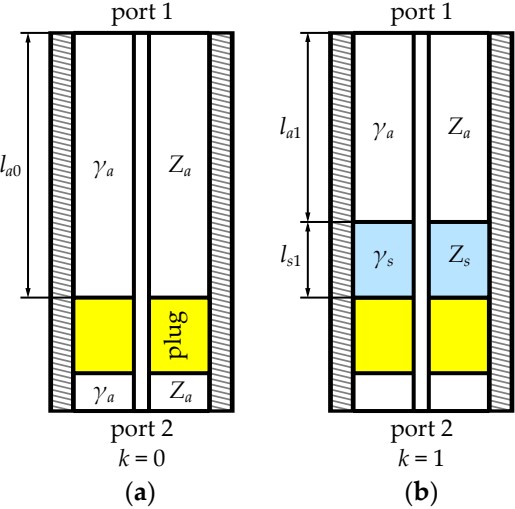

**Figure 2.** Sketch of the semi-open coaxial fixture in two measurement states: (**a**) the empty cell; (**b**) the cell filled with the ring sample of liquid with the ideally flat upper surface. List of symbols: $l_{ak}$—the length of $k$-th airline section; $l_{s1}$—the height of the sample; $Z_a$, $Z_s$—characteristic impedances and $\gamma_a$, $\gamma_s$—propagation constants for the air and sample, respectively.

### 2.1. Classical Approach

In the classical approach, the liquid sample is assumed to be a symmetrical ring with ideally flat surfaces. Transforming (6) to the **S**-matrix notation using (1) we get $\mathbf{S}_{c1}$. Since $\mathbf{S}_{c1}$ describes the sample of liquid whose surface at port 1 is shifted by an unknown length $l_{a1}$ of the airline section, the ratio of the reflection coefficients of $\mathbf{S}_{c1}$ is expressed as follows:

$$\frac{S_{22c}}{S_{11c}} = e^{2\gamma_a l_{a1}} .\tag{7}$$

Then, at each frequency, the length of the airline section and the sample height can be calculated from [15,16]:

$$l_{a1}(f) = \frac{1}{2\gamma_a} \ln \frac{S_{22c}}{S_{11c}} , \qquad l_{s1}(f) = l_{a0} - l_{a1}(f) .\tag{8}$$

from which the optimal values of $l_{a1}$, $l_{s1}$ can be estimated e.g., by the median. That allows the de-embedding of the S-parameters of the sample referenced to the airline impedance:

$$\mathbf{S}_{s1} = \begin{bmatrix} S_{11c}e^{2\gamma_a l_{a1}} & S_{12c}e^{\gamma_a l_{a1}} \\ S_{21c}e^{\gamma_a l_{a1}} & S_{22c} \end{bmatrix}\tag{9}$$

and apply the well-known NRW algorithm [26,27] to obtain the reflection coefficient at the air-sample boundary

$$\Gamma_s = X \pm \sqrt{X^2 - 1} , \qquad X = \frac{S_{11s}^2 - S_{21s}^2 + 1}{2S_{11s}}\tag{10}$$

and the propagation constant

$$\gamma_s = \frac{\ln P}{l_{sk}} , \qquad P = e^{\gamma_s l_{sk}} = \frac{S_{11s} + S_{21s} - \Gamma_s}{1 - \Gamma_s(S_{11s} + S_{21s})} .\tag{11}$$

Although the permittivity of a nonmagnetic liquid can be obtained from both $\Gamma_s$ and $\gamma_s$, the recent one is less prone to the errors of the S-parameter measurement and inexact knowledge of $Z_a$, and thus permittivity calculation from the propagation constant provides more reliable results. For that reason, in this paper we focus on determining $\varepsilon_r$ from $\gamma_s$

$$\varepsilon_r = -\left(\frac{v_a \gamma_s}{2\pi f}\right)^2 ,\tag{12}$$

where $v_a$ is the speed of light in air.

### 2.2. Meniscus Removal Method

As the surface of a liquid in a semi-open cell is not ideally flat, but distorted with a meniscus, the measured sample generally becomes asymmetrical. Therefore, we have proposed a new method dealing with the asymmetry of the distorted sample in [25]. It employs three measurement states: the empty cell and the cell with two different volumes of a liquid for which we assume the reproducible shape of the meniscus, as presented in Figure 3. With the use of **S** matrices measured for all three states, we could de-embed that additional portion of the liquid poured, which is ideally symmetrical. The de-embedded **T**-matrix of that increment volume of the liquid allows calculating both the height increment and the sample permittivity.

Since the height increment of the liquid sample results in the same length decrement of the airline section

$$\Delta l = l_{s2} - l_{s1} = l_{a1} - l_{a2} ,\tag{13}$$

the measurements of two liquid states can be expressed as

$$\mathbf{T}_{c1} = \mathbf{T}_{a1}\mathbf{T}_{s1} = \mathbf{T}_{a2}\mathbf{T}_{a\Delta}\mathbf{T}_{s1} ,\tag{14}$$

$$\mathbf{T}_{c2} = \mathbf{T}_{a2}\mathbf{T}_{s2} = \mathbf{T}_{a2}\mathbf{T}_{s1}\mathbf{T}_{s\Delta} ,\tag{15}$$

where $\mathbf{T}_{a\Delta}$ and $\mathbf{T}_{s\Delta}$ represent the air and liquid sample line sections, respectively, with the same $\Delta l$ length. The first one is described by (4) for $l_{ak} = \Delta l$, while the other, referenced to $Z_a$, is

$$\mathbf{T}_{s\Delta} = \mathbf{T}_t \begin{bmatrix} e^{-\gamma_s \Delta l} & 0 \\ 0 & e^{\gamma_s \Delta l} \end{bmatrix} \mathbf{T}_t^{-1} . \tag{16}$$

Compiling (14) and (15) we get

$$\mathbf{T}_{s\Delta} = \mathbf{T}_{c1}^{-1} \mathbf{T}_{a\Delta} \mathbf{T}_{c2} . \tag{17}$$

In contrast to $\mathbf{T}_{s1}$, $\mathbf{T}_{s\Delta}$ is symmetrical, for not being distorted by the meniscus, therefore, from the symmetry condition $T_{12} = -T_{21}$ in $\mathbf{T}$-matrix notation, we determine the increment height of the sample, by calculating the median value of

$$\Delta l(f) = \frac{\ln r}{2\gamma_a} , \qquad r = e^{2\gamma_a \Delta l} = \frac{T_{22c}^{(1)} T_{12c}^{(2)} - T_{21c}^{(1)} T_{11c}^{(2)}}{T_{12c}^{(1)} T_{22c}^{(2)} - T_{11c}^{(1)} T_{21c}^{(2)}} , \tag{18}$$

where $T_{ijc}^{(k)}$ are the adequate parameters of $\mathbf{T}_{ck}$.

Then, after determining the trace of (16) and (17)

$$\mathrm{tr}\,\mathbf{T}_{s\Delta} = e^{-\gamma_s \Delta l} + e^{\gamma_s \Delta l} = 2\cosh \gamma_s \Delta l , \tag{19}$$

we calculate the propagation constant of the liquid sample

$$\gamma_s = \frac{1}{\Delta l}\mathrm{arcosh}\left(\frac{1}{2}\mathrm{tr}\,\mathbf{T}_{s\Delta}\right) , \tag{20}$$

that is not disturbed by the meniscus. The permittivity is then determined from (12).

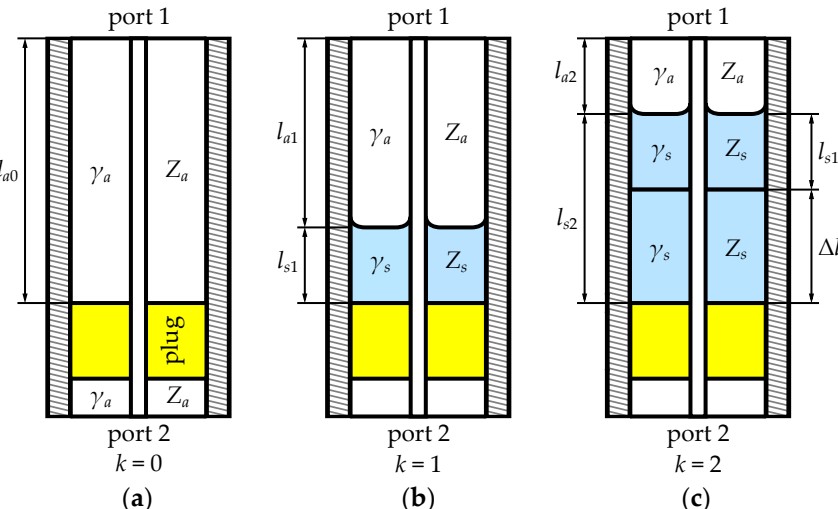

**Figure 3.** Sketch of the semi-open coaxial fixture in its three measurement states of liquid under test distorted by the meniscus: (**a**) the empty cell; (**b**) the initial volume of a liquid; (**c**) the final volume. List of symbols: $l_{ak}$—the length of $k$-th airline section; $l_{sk}$—the height of $k$-th sample; $\Delta l$—the height increment; $Z_a$, $Z_s$—characteristic impedances and $\gamma_a$, $\gamma_s$—propagation constants for the air and sample, respectively.

## 3. EM Simulation

In contrast to the real VNA measurements, the EM simulations allow determining the S-parameters of the fixture measurement states with an uncertainty that can be kept low enough by establishing appropriate simulation conditions. Therefore, such EM results

allow, in a controlled way, a thorough investigation of effects caused by various surface distortions and determining pertinent errors of the permittivity calculations.

For this goal, we simulated the measurements performed in our 7 mm coaxial fixture employed in [16,25], as shown in Figure 4, at frequencies up to 18 GHz. The fixture was filled with vacuum, as a good substitute for air, and with the liquid sample (the blue layer). The dielectric plug (the yellow layer), that keeps the liquid in place in real measurements, was simulated as PTFE with constant relative permittivity $\varepsilon_r = 2.1$. The inner conductor was defined as the perfect electric conductor (PEC) as well as the vertical boundaries of the vacuum and liquid sample performing as the outer conductor.

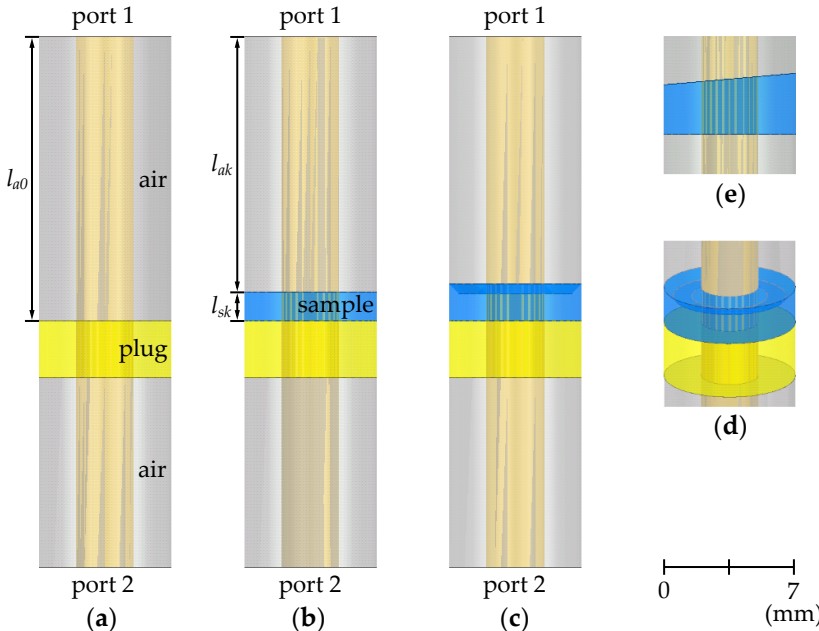

**Figure 4.** The simulated semi-open coaxial test fixture. Cases: (**a**) the empty cell, (**b**) the ideal sample of liquid with the flat upper surface, (**c**) the sample with the meniscus; (**d**) another view of the meniscus under analysis; (**e**) the tilted sample.

To study how the meniscus affects the permittivity results we simulated the empty fixture and the fixture in four states per liquid tested. The first two were the cases for the sample with the meniscus, the initial and the final volumes of liquid. Two remaining ones were the cases for the ideal samples, with the flat and transversal surfaces of the same volumes of the liquid as the ones with the meniscus. The height of the final volume of liquid was twice the initial ones so that the increment volume of liquid had the same height as the first state. We arbitrarily shaped the meniscus as a cylinder with a truncated cone cut out, shown in Figure 4c,d. As the real shape of the meniscus is somewhat different, the conclusions from this analysis should be treated as rather qualitative, not quantitative.

We also studied the effects of the not perfectly vertical position of the fixture as another type of liquid surface distortions. To this end, we simulated the skewed samples, shown in Figure 4e, with 1° and 5° tilts. The first one is a reasonable value that may occur during measurements, and the second one is rather too high to be unnoticeable, but helps to observe the trend. The dielectric plug is omitted for simplicity in this test.

We examined three liquids of different permittivity: relatively low-loss propan-2-ol (IPA) [29], medium-loss 50% aqueous solution of IPA (IPA50%) [25] and high-loss distilled water [30], to explore how the errors depend on the liquid properties. The reference permittivity of each of them, necessary for the simulations, was determined according to data available in the just mentioned references.

The EM simulations were performed with ANSYS®HFSS, 3D finite element method frequency-domain solver [31]. The excitations were set using the wave ports, thus the port

fields were defined exactly with no need for an additional port calibration, according to [32]. The order of basis functions that HFSS uses to interpolate field values from the nodal values was set to "mixed". The adaptive meshing was broadband with 30% maximum refinement per pass. A special curvilinear meshing was applied to all curved surfaces.

Several simulations of the fixture filled with the ideal sample of water were performed to estimate the required accuracy—parameter Delta S in HFSS, which is the maximum difference between **S**-matrices determined for the consecutive passes. The data for different setups are presented in Table 1. The results were compared with the mathematical model, presented in Figure 5. The relative error of relevant complex value $z \in \{S_{11}, S_{21}, \varepsilon_r\}$ is calculated as $\delta z = \mathrm{abs}\left(\frac{z_{simulated} - z_{exact}}{z_{exact}}\right) \times 100\%$. For the permittivity errors less than 0.1% the reasonable value of Delta S is $10^{-5}$. The maximum number of passes was set experimentally to 25, regarding the available computing unit with 64 GB RAM. The maximum change of the port impedance between the passes was tested as not critical and set to 0.2% (parameter Delta Z0).

**Table 1.** Juxtaposition of the simulation parameters for different values of accuracy (the parameter Delta S) set—the number of passes to converge, the computation time, the number of simulated tetrahedra and the achieved accuracy.

| Delta S Set | Pass | Computation Time | Tetrahedra | Delta S Achieved |
|:-----------:|:----:|:----------------:|:----------:|:----------------:|
| $10^{-3}$ | 6 | 4 min | 5.6 k | $4 \times 10^{-4}$ |
| $10^{-4}$ | 10 | 7 min | 13 k | $6 \times 10^{-5}$ |
| $10^{-5}$ | 21 | 7 h | 212 k | $7 \times 10^{-6}$ |

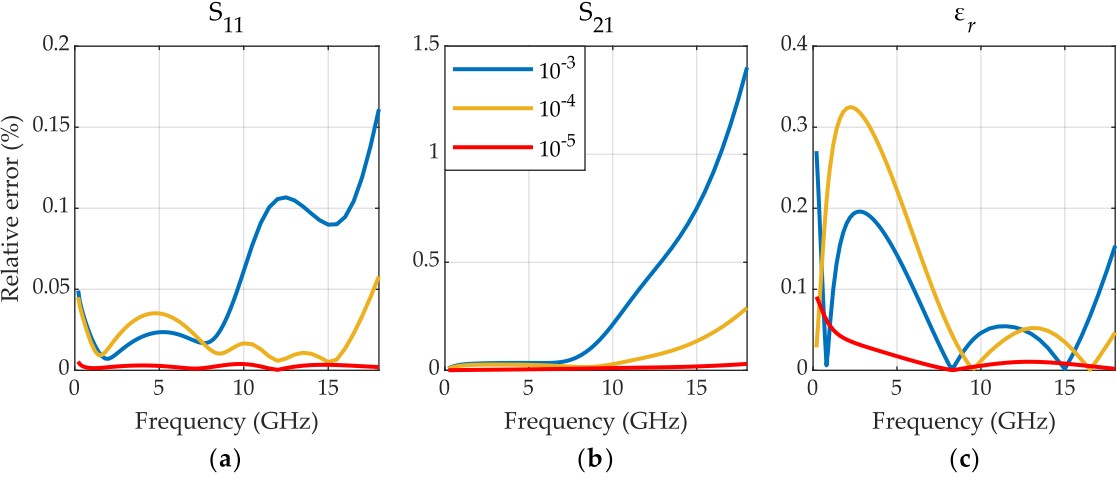

**Figure 5.** The relative error of (**a**) $S_{11}$, (**b**) $S_{21}$ and (**c**) $\varepsilon_r$ for the line filled with 2 mm column of water for the different values of the accuracy represented by the parameter Delta S [31].

## 4. Simulation Results

In Section 4.1 we analyze errors in the permittivity and height determination for the samples distorted with the meniscus and then, in Section 4.2 we present a similar analysis for the tilted test cell.

### 4.1. Analysis of Errors Caused by the Meniscus

Permittivity characteristics determined with the NRW and the MR methods for IPA, IPA50% and distilled water are shown in Figure 6a–c, respectively. The characteristics for samples with the flat surface are presented with solid lines while dashed-dotted lines introduce the results for the samples distorted with the meniscus. The characteristics of the reference permittivity, used by the EM simulator, are illustrated with black lines that

are, however, completely covered by the other lines. Computations of simulations lasted, depending on the case, from 2 to 50 h with 100–750 thousand tetrahedra used.

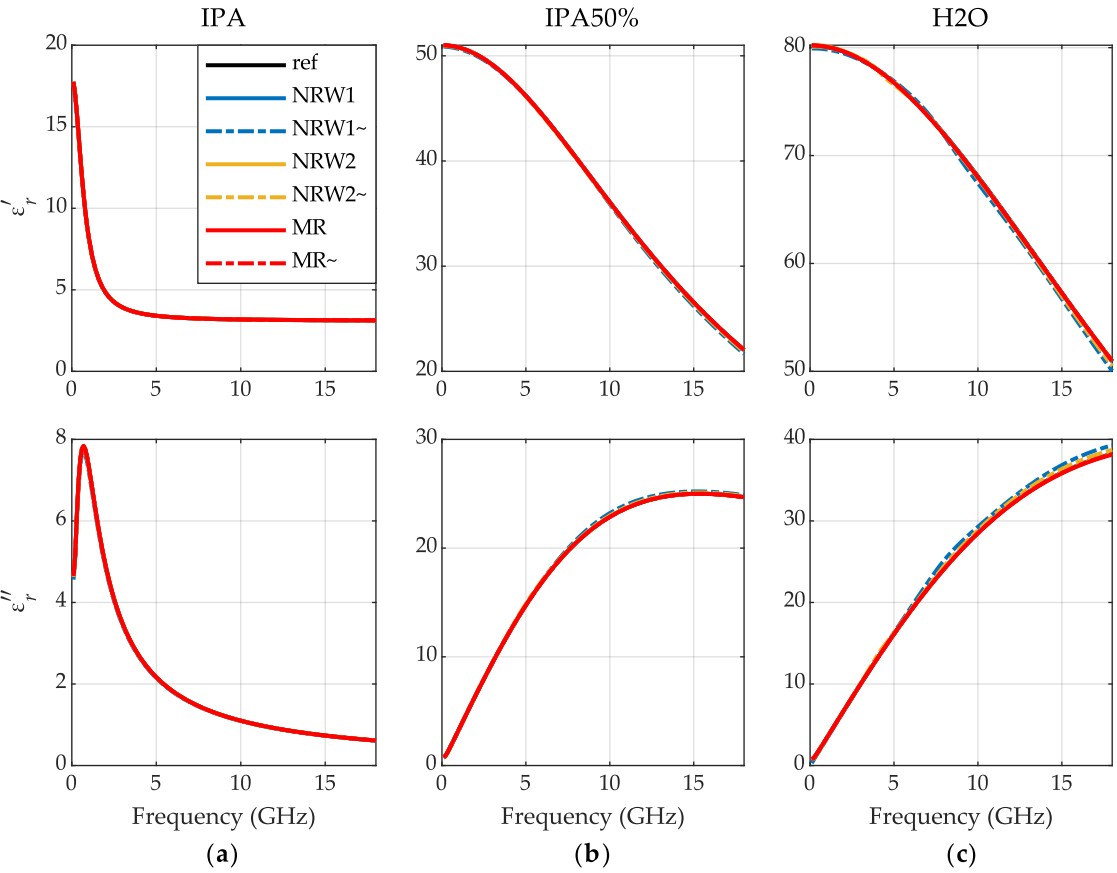

**Figure 6.** Characteristics of the real ($\varepsilon'_r$) and imaginary part ($\varepsilon''_r$) of permittivity for (**a**) IPA, (**b**) 50% aqueous solution of IPA, (**c**) distilled water. Determined from 0.1 to 18 GHz. Line types: solid—the ideal sample with the flat surface, dashed-dotted—the sample with the meniscus (denoted by ~). Line colors: black—reference, red—the outcome of the meniscus removal (MR) method, blue and yellow—the NRW method for the initial and the final volumes of liquid, respectively.

All the characteristics for IPA, which has the lowest permittivity of the three liquids, overlap regardless of the sample shape, methods applied and volume of the liquid. For samples with the meniscus, as the permittivity increases, we observe slight deviations from the reference value at higher frequencies.

The relative errors of the sample column height versus frequency with Somlo's and the MR methods are presented in Figure 7. For frequencies below 1 GHz, the error traces tend to deviate from 0, due to a finite angular resolution of the simulated S-parameters. Except this region, the errors for the ideal samples deviate by less than 0.3% for all the liquids, levels and methods. Since the errors of the ideal samples should be 0, such small deviations obtained confirm the high accuracy of the performed simulations. The estimated values of sample column heights used for the permittivity calculations and their relevant errors are shown in Table 2. Thanks to the broadband robust approach, individual large error deviations seen in Figure 7 hardly impact the estimated column heights.

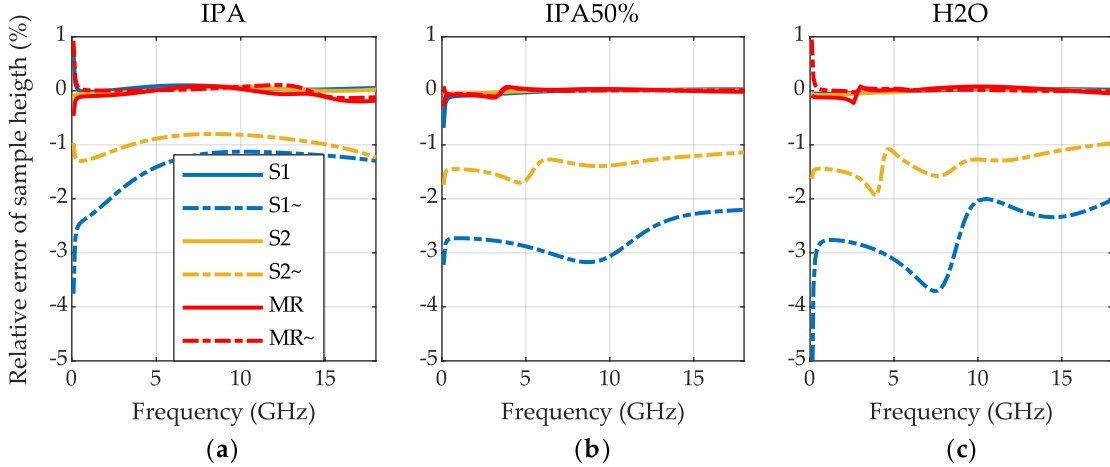

**Figure 7.** The relative error of sample height versus frequency determination for (**a**) IPA, (**b**) 50% aqueous solution of IPA and (**c**) distilled water. Line types: solid—the ideal sample with the flat surface, dashed-dotted—the sample with the meniscus (denoted by ~). Line colors: red—the outcome of the meniscus removal (MR) method, blue and yellow—Somlo's method [15] for the initial (S1) and the final (S2) volumes of liquid, respectively.

**Table 2.** Juxtaposition of the simulated and estimated column heights of samples $l_s$ with their relative errors determined for different liquids, sample surface types (the flat and with the meniscus) and the methods used: S1, S2—Somlo's method for the initial and final volumes of liquid, respectively, MR—the meniscus removal method for the incremental column height.

| Liquid | | IPA | | | IPA50% | | | H2O | | |
|---|---|---|---|---|---|---|---|---|---|---|
| **Sample** | **Method** | **S1** | **S2** | **MR** | **S1** | **S2** | **MR** | **S1** | **S2** | **MR** |
| Flat surface | Simulated $l_s$ [mm] | 2.000 | 4.000 | 2.000 | 2.000 | 4.000 | 2.000 | 2.000 | 4.000 | 2.000 |
| | Estimated $l_s$ [mm] | 2.001 | 4.001 | 1.999 | 2.000 | 4.000 | 2.000 | 2.000 | 4.000 | 2.001 |
| | Relative error [%] | 0.04 | 0.02 | −0.05 | 0.01 | 0.01 | 0.02 | 0.02 | 0.01 | 0.03 |
| Meniscus | Simulated $l_s$ [mm] | 2.000 | 4.000 | 2.000 | 2.000 | 4.000 | 2.000 | 2.000 | 4.000 | 2.000 |
| | Estimated $l_s$ [mm] | 1.976 | 3.963 | 2.000 | 1.945 | 3.946 | 2.000 | 1.949 | 3.949 | 2.000 |
| | Relative error [%] | −1.22 | −0.92 | 0.00 | −2.76 | −1.34 | 0.01 | −2.54 | −1.29 | 0.01 |

The results for the samples with the meniscus introduce clearly visible differences. The heights calculated with Somlo's method are lower than expected, while the errors of the MR method—presented with red curves—stay small, below 0.2% in the whole frequency range.

Small differences in the permittivity characteristics shown in Figure 6 can be better perceived as the relative errors that are presented in Figure 8. For the ideal samples with the flat surface, presented with continuous lines, the errors, as one can expect, are negligible. On the other hand, the results for samples with the meniscus obtained with the NRW algorithm are burdened with the error. We can formulate general conclusions, the higher permittivity, the more significant errors in the permittivity determination, as the wavelength is smaller at the same frequency and surface distortion affects S-parameters more. Also, lower samples introduce bigger errors because the meniscus is a relatively larger distortion for them.

The permittivity results obtained with the MR algorithm are highly consistent over the whole frequency band. The errors calculated for the ideal and the distorted samples as well do not exceed 0.3% for all the liquids. This means that according to its purpose, the MR method is robust to the liquid surface distortions caused by a meniscus. The errors achieved with the NRW method are of an order of magnitude bigger. The above conclusions remain the same for the convex meniscus, also examined.

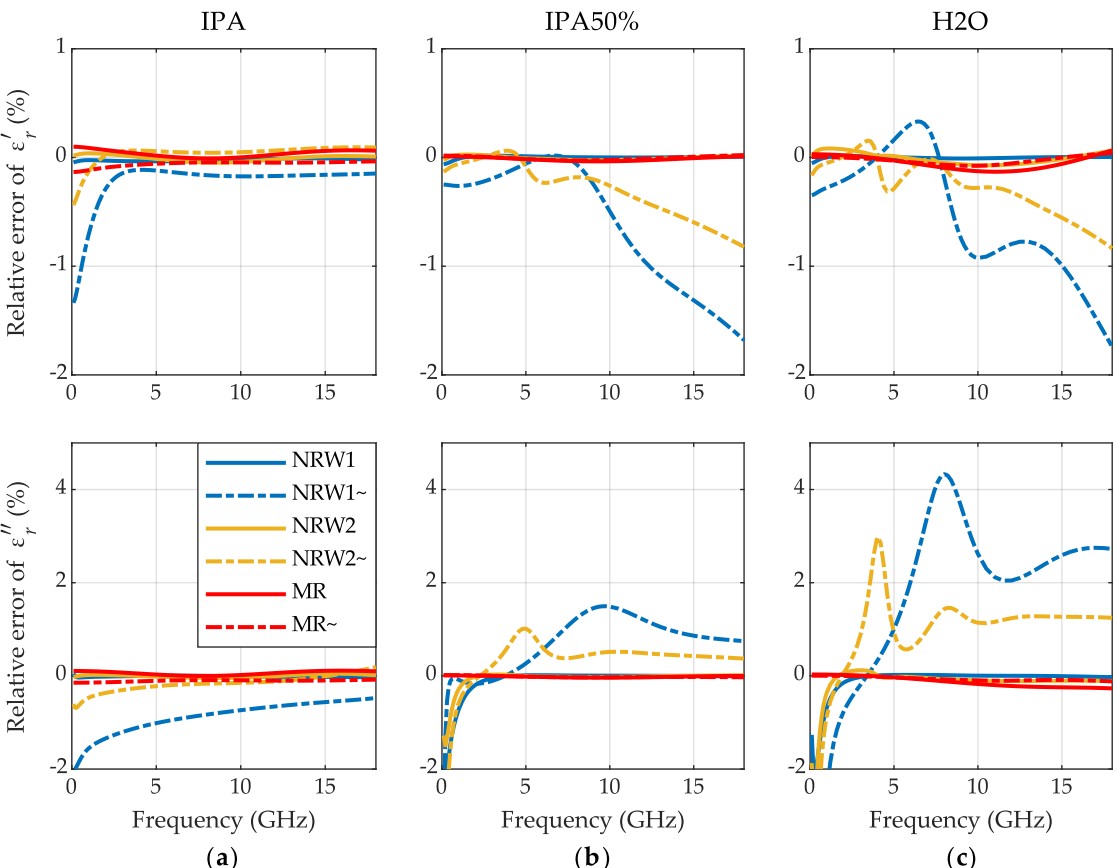

**Figure 8.** The relative error of the real ($\varepsilon'_r$) and imaginary ($\varepsilon''_r$) part of permittivity for (**a**) IPA, (**b**) 50% aqueous solution of IPA, (**c**) distilled water. Line types: solid—the ideal sample with the flat surface, dashed-dotted—the sample with the meniscus (denoted by ~). Line colors: red—the outcome of the meniscus removal (MR) method, blue and yellow—the NRW method for the initial and the final volumes of liquid, respectively.

### 4.2. Analysis of Errors Caused by Tilt (Skew) of the Cell

In this subsection we compare the errors of the permittivity determination for ideal transversal ring samples of water and samples tilted by 1° and 5°. For each case two volumes of liquid were simulated. The dielectric plug was omitted for simplicity. The simulation calculations lasted from 15 to 40 h and stopped not converged on the last, 25th, adaptive pass with Delta S about $10^{-3}$ simulating from 400 to 750 thousand tetrahedra.

The heights determined for the tilted samples at each frequency show ripples that can be observed in Figure 9 with relative errors. For 1° of tilt, the MR method seems to be more sensitive than the NRW. Nevertheless, despite quite big errors visible on the plots, the estimated optimal heights, given in Table 3, stay almost perfect. This confirms the great advantage of broadband measurements and robust estimation of the sample height, which allows the elimination of relatively big errors at particular frequencies.

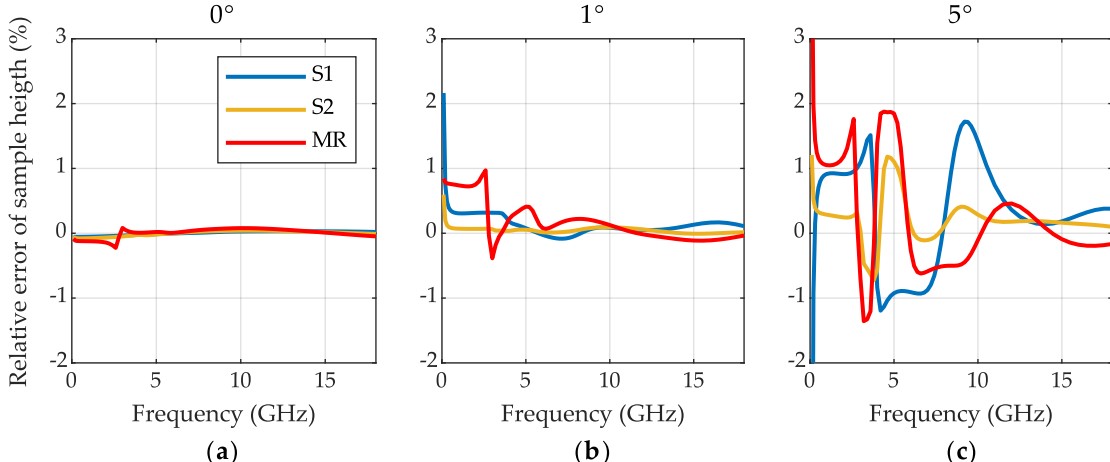

**Figure 9.** The relative error of sample column height calculation versus frequency for distilled water for (**a**) the perfectly vertical cell position, (**b**) 1° and (**c**) 5° tilted cell. Line colors: red—the outcome of the meniscus removal (MR) method, blue and yellow—Somlo's method [15] for the initial (S1) and the final (S2) volumes of liquid, respectively.

**Table 3.** Juxtaposition of simulated and estimated column heights of samples $l_s$ with their relative errors determined for distilled water for the different tilt of the cell and the methods used: S1, S2—Somlo's method for the initial and final volumes of liquid, respectively, MR—the meniscus removal method for the incremental column height of the liquid.

| Tilt Angle | 0° | | | 1° | | | 5° | | |
|---|---|---|---|---|---|---|---|---|---|
| Method | S1 | S2 | MR | S1 | S2 | MR | S1 | S2 | MR |
| Simulated $l_s$ [mm] | 2.000 | 4.000 | 2.000 | 2.000 | 4.000 | 2.000 | 2.000 | 4.000 | 2.000 |
| Estimated $l_s$ [mm] | 2.000 | 4.000 | 2.001 | 2.002 | 4.002 | 2.000 | 2.007 | 4.007 | 1.998 |
| Relative error [%] | 0.02 | 0.01 | 0.03 | 0.09 | 0.04 | 0.02 | 0.35 | 0.18 | −0.10 |

The relative error of the water permittivity is presented in Figure 10. For small tilt the performance of the NRW and the MR methods are similar. For higher tilt, the MR algorithm is better for higher-frequency end, but at lower frequencies introduces bigger ripples and spikes. We have also examined tilted samples of IPA, which has lower permittivity. The results more closely adhered to the reference data than for water. Similarly, as with the meniscus, the higher permittivity, the higher sensitivity to the tilt distortions is observed.

Despite the MR method eliminates the effects caused by reproducible surface distortions by de-embedding the symmetrical increment volume of the liquid sample from two states, in the case of tilted samples the permittivity results are clearly more perturbed at the higher tilt—Figure 10c. Due to the skew of the sample surface, the higher field modes appear, which at higher frequencies freely propagate in the sample. In the liquid, the EM wave slows down, and in the place where the sample is higher, the phase is more delayed. It is manifested by the asymmetry of the EM field distribution in the vicinity of a tilted sample, visible in Figure 11b. The propagation of the higher modes has not been accounted for in the MR nor other known methods. Outside the sample, those modes cannot propagate in the air-filled 7 mm coaxial line, and thus, at the ends of the picture, the field distribution becomes symmetrical again before reaching the ports. The propagation of the wave through the sample was disturbed and the errors could not be eliminated, especially for bigger tilt. In the samples with the meniscus—Figure 11a—higher modes also appear, but due to the symmetry of the distortion, the field distribution stays symmetrical. The above results show, for the first time, how important is the vertical adjustment of the cell position within 1° to provide appropriate conditions for accurate permittivity measurements.

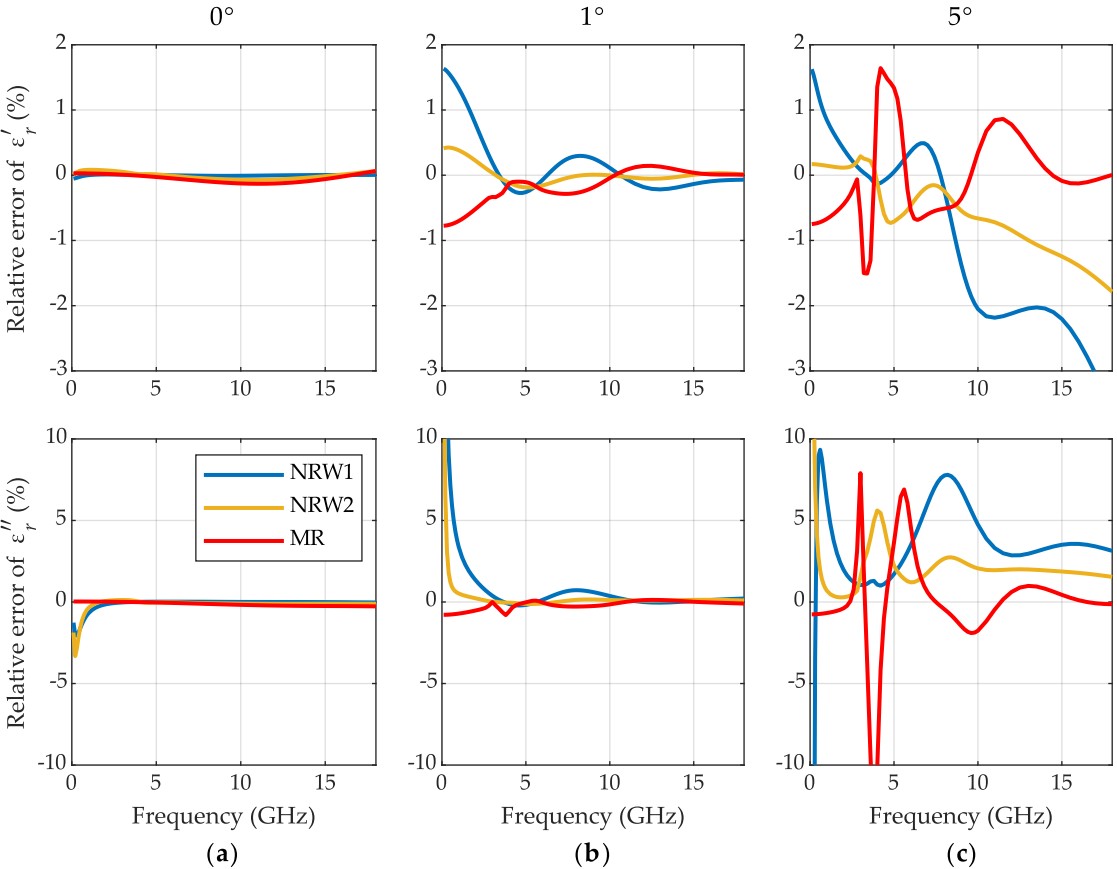

**Figure 10.** The relative error of the real ($\varepsilon'_r$) and imaginary ($\varepsilon''_r$) part of permittivity for distilled water for (**a**) the perfectly vertical cell position, (**b**) 1° and (**c**) 5° tilted cell. Line colors: red—the outcome of the meniscus removal (MR) method, blue and yellow—the NRW method for the initial and the final volumes of liquid, respectively.

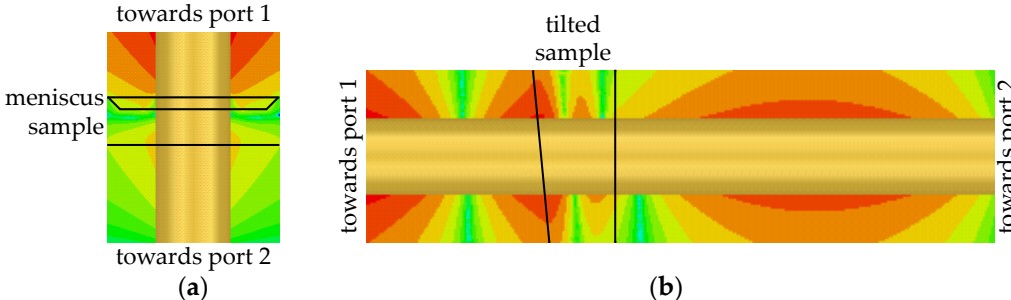

**Figure 11.** Fragment of the electric field distribution in the semi-open coaxial test fixture filled with a volume of distilled water simulated in HFSS. (**a**) The sample with the meniscus, (**b**) the sample tilted by 5° (rotated view). The dielectric plug is omitted for simplicity.

## 5. Conclusions

We have studied how surface distortions of liquid samples measured in a semi-open coaxial cell affect a broadband permittivity determination due to a meniscus and a deviation from the strictly vertical position of the cell. To this end, we performed EM simulations of the two-port scattering parameter measurements of samples in the 7 mm cell up to 18 GHz with the distorted surface as well as the ideal ones, i.e., being flat and transversal, which served as the reference. From the scattering matrices obtained, we calculated the permittivity characteristics of the liquid samples using the classic NRW algorithm [26,27] and our new method [25], which is capable of meniscus removal.

Our study shows that the meniscus affects the S-parameters at the high-frequency end and its impact is greater for the liquid of higher permittivity. Consequently, the errors in

the permittivity determination with the classic NRW method cannot be ignored, while our MR method stays robust to the reproducible meniscus and provides smaller errors.

However, none of the verified methods stays resistant to the errors caused by the tilt of the test cell. The MR technique, despite the mathematical removal of the top layer of the liquid, introduces significant errors especially when the tilt and permittivity are bigger. The axial asymmetry of the field distribution causes effects not accounted for in the model describing the measurements. Therefore, a precise vertical adjustment of the cell is necessary to maintain the errors small enough.

Although the above conclusions are at present rather qualitative, they may be treated as general guidelines for providing high accuracy results of the permittivity measurements performed with the semi-open cells. We plan to enhance this study in the future by simulating the real shape of the meniscus.

**Author Contributions:** Conceptualization, W.W. and M.K.; methodology, M.K. and W.W.; software, M.K.; validation, W.W.; formal analysis, M.K. and W.W.; investigation, M.K. and W.W.; data curation, M.K.; writing—original draft preparation, M.K.; writing—review and editing, M.K. and W.W.; visualization, M.K.; supervision, W.W.; project administration, W.W. All authors have read and agreed to the published version of the manuscript.

**Funding:** This research received external funding by the National Science Centre, Poland—grant No. 2020/37/N/ST7/04046, program PRELUDIUM.

**Institutional Review Board Statement:** Not applicable.

**Informed Consent Statement:** Not applicable.

**Data Availability Statement:** The data presented in this paper are available on request from the corresponding author.

**Acknowledgments:** We thank Adam Abramowicz, Arkadiusz Lewandowski and Mateusz Żbik for their valuable advice regarding the EM simulation software.

**Conflicts of Interest:** The authors declare no conflict of interest. The funders had no role in the design of the study; in the collection, analyses, or interpretation of data; in the writing of the manuscript, or in the decision to publish the results.

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
