# Peer review of "Errors in Broadband Permittivity Determination Due to Liquid Surface Distortions in Semi-Open Test Cell"

_remotesensing, doi:10.3390/rs13050983_

Round 1

Reviewer 1 Report

Kalisiak and Wiatr present a very technical and detailed study about permittivity characterizations in microwave resonators. Overall, I think their paper is suitable for publication once the following comments have been satisfactorily addressed. In general, they do not provide enough details for the reader to understand the manuscript without constantly referring to references.

  1. Line 15: The reader needs more background on existing techniques to determine a liquid’s permittivity. First, please briefly summarize techniques not based on microwaves. Second, a bit more discussion about microwave-based techniques is needed. In particular, a few references to unconventional microwave setups that promise to boost the precision is missing. These are
    • Self-oscillating sources, like in COHEN, Seth D., CAVALCANTE, Hugo LD de S., GAUTHIER, Daniel J. Subwavelength position sensing using nonlinear feedback and wave chaos. Physical Review Letters, 2011, vol. 107, no 25, p. 254103.
    • Perfectly absorbing scattering systems with a diverging time delay, like in F. IMANI, Mohammadreza, SMITH, David R., DEL HOUGNE, Philipp. Perfect Absorption in a Disordered Medium with Programmable Meta‐Atom Inclusions. Advanced Functional Materials, 2020, vol. 30, no 52, p. 2005310.
    • A multi-port version of the latter, together with theory on why the time delay enhances the sensitivity can be found in arXiv:2010.06438.
  2. Line 29: “a new method” please briefly describe the method for the reader’s benefit so that one does not have to look up the reference.
  3. Lines 32-33: “conventional techniques” please briefly describe these for the reader’s benefit.
  4. Figure 1: Please add a photo to illustrate the meniscus effect in real life.
  5. End of page 2: to determine the height of the sample, would it be possible to inverse Fourier transform the spectrum to see when the first pulse is reflected and determine the height form the time of flight?
  6. Line 57: “NRW algorithm” please briefly explain this algorithm for the benefit of the reader.
  7. Simulation setup: please provide details on the solver (besides stating the software name).
  8. Simulation setup: please explain how the ports are simulated. Are they realistic?
  9. Section 4: please provide a summary including mathematical details of the different methods that are compared for the benefit of the reader.

Author Response

We would like to thank you for your detailed review that has helped us to improve the paper.

The changes resulting from reviewer’s 1 comments are highlighted in yellow in uploaded “paper17_differences.pdf” file.

The changes resulting from reviewer’s 3 comments are highlighted in blue in uploaded “paper17_differences.pdf” file.

The changes resulting from reviewer’s 1 and 3 comments are highlighted in green in uploaded “paper17_differences.pdf” file.

Another changes are marked by blue text color.

Point 0: Kalisiak and Wiatr present a very technical and detailed study about permittivity characterizations in microwave resonators. Overall, I think their paper is suitable for publication once the following comments have been satisfactorily addressed. In general, they do not provide enough details for the reader to understand the manuscript without constantly referring to references.

Response 0: Thank you very much for your review, we have tried to include missing information for the reader's convenience.

Point 1: Line 15: The reader needs more background on existing techniques to determine a liquid’s permittivity. First, please briefly summarize techniques not based on microwaves. Second, a bit more discussion about microwave-based techniques is needed. In particular, a few references to unconventional microwave setups that promise to boost the precision is missing. These are

  • Self-oscillating sources, like in COHEN, Seth D., CAVALCANTE, Hugo LD de S., GAUTHIER, Daniel J. Subwavelength position sensing using nonlinear feedback and wave chaos. Physical Review Letters, 2011, vol. 107, no 25, p. 254103.
  • Perfectly absorbing scattering systems with a diverging time delay, like in F. IMANI, Mohammadreza, SMITH, David R., DEL HOUGNE, Philipp. Perfect Absorption in a Disordered Medium with Programmable Meta‐Atom Inclusions. Advanced Functional Materials, 2020, vol. 30, no 52, p. 2005310.
  • A multi-port version of the latter, together with theory on why the time delay enhances the sensitivity can be found in arXiv:2010.06438.

Response 1: We have added additional information on existing techniques in a broad frequency range with relative references (lines 15-17). Thank you for the references, however, the revision of not conventional methods is, in our opinion, out of the scope of this publication, being the topic for a separate paper.

Point 2: Line 29: “a new method” please briefly describe the method for the reader’s benefit so that one does not have to look up the reference.

Response 2: We agree with this remark. Section 2 has been thoroughly rewritten to comprise more detailed information about the meniscus removal method.

Point 3: Lines 32-33: “conventional techniques” please briefly describe these for the reader’s benefit.

Response 3: In rebuilt Section 2 there has been also added description of NRW and Somlo’s methods used.

Point 4: Figure 1: Please add a photo to illustrate the meniscus effect in real life.

Response 4: We have added such a photo (Figure 1).

Point 5: End of page 2: to determine the height of the sample, would it be possible to inverse Fourier transform the spectrum to see when the first pulse is reflected and determine the height form the time of flight?

Response 5: Such measurements should be possible, VNA can be used as a TDR (with low noise due to narrow RF bandwidth), according to [https://www.academia.edu/5764136/TDR_and_VNA_Measurement_Primer]. However, we did not test such the solution and its resolution, accuracy. It is not the subject of this paper.

Point 6: Line 57: “NRW algorithm” please briefly explain this algorithm for the benefit of the reader.

Response 6: As in point 2.

Point 7: Simulation setup: please provide details on the solver (besides stating the software name).

Response 7: We have added detailed information about simulations in Section 3 – lines 142-154.

Point 8: Simulation setup: please explain how the ports are simulated. Are they realistic?

Response 8: Yes, ports are realistic, we have added such information in lines 143-144, based on [K. Wu, et al. The Match Game: Numerical De-Embedding of Field Simulation and Parameter Extraction of Circuit Models for Electromagnetic Structure Using Calibration Techniques].

Point 9: Section 4: please provide a summary including mathematical details of the different methods that are compared for the benefit of the reader.

Response 9: In section 2 we have added also mathematical details that form the used algorithms.

Thank you again for your review.

Reviewer 2 Report

In the paper titled " Errors in Broadband Permittivity Determination due
to Liquid Surface Distortions in Semi-Open Test Cell
", the authors have provided a nice piece of study corresponding to the generation of errors in broadband permittivity determination due to presence of meniscus and tilt of semi-open coaxial test cell based on scattering parameters, obtained using the electromagnetic simulations of samples in a broad frequency range up to 18 GHz. In numerical analysis, the authors have shown that the estimated permittivity error can be minimized by using the MR method.

As a technical note, it appears fine to accept the paper, as the authors have already provided the details of the method in their earlier works [14], [17]. A numerical study is presented in this work, whereas the authors already develop the method. Hence, as a technical note, the current manuscript may be acceptable.

Author Response

We would like to thank you for the positive reception of our paper and your review.

Reviewer 3 Report

Since the analysis of the performance of the MR method (and comparison with NRW method) is based in numerical simulations, it is important to give information about those simulations: method (I assume FEM because the software is Ansys-HFSS), # divisions per wavelength, total # of elements or unknowns (edges), computational time (in a given CPU), use of adaptive meshing or not…).

Moreover, it would be interesting to estimate the error (accuracy) due to the EM simulation. Authors could launch a set of simulations refining the mesh in order to estimate the error due to the mesh density used for these results.

Another permittivity measurement technique for liquids is the dielectric probe, a one-port (reflection) technique without the issue of the meniscus, since the liquid is completely in contact with the probe (coaxial port). It would be interesting to compare the transmission/reflection meniscus removal method with the dielectric probe method.

Why the reference value of permittivity (dielectric constant and loss factor) in fig. 2 is not exactly the same than in [14]. Is it because of a different temperature (30ºC in [14])? Authors should explain it.

In line 73, 76, I assume it is “air” instead of “vacuum”. Please correct if I am right.

Author Response

We would like to thank you for your detailed review that has helped us to improve the paper.

The changes resulting from reviewer’s 1 comments are highlighted in yellow in uploaded “paper17_differences.pdf” file.

The changes resulting from reviewer’s 3 comments are highlighted in blue in uploaded “paper17_differences.pdf” file.

The changes resulting from reviewer’s 1 and 3 comments are highlighted in green in uploaded “paper17_differences.pdf” file.

Another changes are marked by blue text color.

Point 1: Since the analysis of the performance of the MR method (and comparison with NRW method) is based in numerical simulations, it is important to give information about those simulations: method (I assume FEM because the software is Ansys-HFSS), # divisions per wavelength, total # of elements or unknowns (edges), computational time (in a given CPU), use of adaptive meshing or not…).

Response 1: We have added detailed information about simulations in Section 3 – lines 142-154.

Point 2: Moreover, it would be interesting to estimate the error (accuracy) due to the EM simulation. Authors could launch a set of simulations refining the mesh in order to estimate the error due to the mesh density used for these results.

Response 2: We did such experiments before running simulations used in the paper, however, we did not published them. Now they have been added, Table 1 and Figure 5 with results for different accuracy and argumentation for setup used.

Point 3: Another permittivity measurement technique for liquids is the dielectric probe, a one-port (reflection) technique without the issue of the meniscus, since the liquid is completely in contact with the probe (coaxial port). It would be interesting to compare the transmission/reflection meniscus removal method with the dielectric probe method.

Response 3: As we understand, you are referring to the dielectric probe, open at the end, immersed in liquid under test. Yes, it would be interesting, however, it is out of the scope of this article.

We have got no experience in such measurements, nevertheless from [F. Costa, et al. Electromagnetic Characterisation of Materials by Using Transmission/Reflection (T/R) Devices]:

“Free-space reflection and open-ended probe techniques are typical transmission/reflection or reflection-only NDT methods [23–28]: they are indeed attractive because they maintain material integrity and are increasingly used in various fields (mechanics, constructions, medicine, and so on), but they usually pay a price in terms of accuracy and mathematical model complexity with respect to the guided T/R methods.

We have added that reference and a brief mention of that method (reflection-only) in the introduction – line 17.

Point 4: Why the reference value of permittivity (dielectric constant and loss factor) in fig. 2 is not exactly the same than in [14]. Is it because of a different temperature (30ºC in [14])? Authors should explain it.

Response 4: It is (on average, because of spikes), referring to measurements carried with the meniscus removal method in the cited article. However, the exact value of the reference permittivity is not important in this paper, but the errors introduced by the algorithms tested.

Point 5: In line 73, 76, I assume it is “air” instead of “vacuum”. Please correct if I am right.

Response 5: There was a certain inconsistency here. We have corrected that part, clarifying that air, which is normally in the cell during measurements, is approximated in simulations by the vacuum – lines 120-121.

Thank you again for your review.

Round 2

Reviewer 1 Report

The authors have made a considerable effort to address the portion of my concerns regarding missing details in their initial manuscript. 

The current introduction is not doing the authors a favor. A good introduction convinces as many readers as possible of the value that the paper has for them. Therefore, it is helpful to place the work within a broad context.

The main problem is that the authors fail to provide a physical intuition of how the permittivity sensing works. What is the underlying mechanism? The new sentence that they added lists a few specific methods (unknown to the non-expert reader) without any description but doesn't help the reader to orientate within the field.

I would assume that most permittivity-sensing methods are based on resonances, which are quite sensitive to changes in permittivity (or any other parameter) because the waves remain for a long time within a system at resonance. The authors must provide such an intuitive explanation (possibly a different one if my intuition is wrong) to help the reader understand what this is all about, before diving into highly technical details. If my assumption about the role of the dwell time is correct, it would naturally make sense to add a sentence about systems that are tailored to display special scattering anomalies (exceptional points, self-oscillation, (coherent) perfect absorption) because these special settings boost the sensitivity even further. In addition to the previously proposed references, I can also recommend:

Chen, W., Özdemir, Ş. K., Zhao, G., Wiersig, J., & Yang, L. (2017). Exceptional points enhance sensing in an optical microcavity. Nature548(7666), 192-196.

and

 https://arxiv.org/abs/2012.10402 

To summarize, a thorough well-written discussion of context is not "beyond our scope" but essential for any good paper.

In addition, there are some language issues that need to be sorted out before publication. I cannot list all problems here, but to give a representative example, here is one:

"To get rid of the meniscus effect authors of [20–22] applied a cell ..."

should read

"To get rid of the meniscus effect, the authors of [20–22] applied a cell ..."

Author Response

We would like to thank the reviewer for the second review that again has helped us to improve the paper and find many language issues.

The changes resulting from the second review report by 1st reviewer are marked by blue text color or highlighted in yellow in the uploaded “paper20_differences.pdf” file.

Point 1: The current introduction is not doing the authors a favor. A good introduction convinces as many readers as possible of the value that the paper has for them. Therefore, it is helpful to place the work within a broad context.

The main problem is that the authors fail to provide a physical intuition of how the permittivity sensing works. What is the underlying mechanism? The new sentence that they added lists a few specific methods (unknown to the non-expert reader) without any description but doesn't help the reader to orientate within the field.

Response 1: We have added a brief description of the physical aspect of permittivity and also a comparison of resonant and broadband two-port microwave measurements, which are the most popular methods for determining the permittivity.

Pont 2: I would assume that most permittivity-sensing methods are based on resonances, which are quite sensitive to changes in permittivity (or any other parameter) because the waves remain for a long time within a system at resonance. The authors must provide such an intuitive explanation (possibly a different one if my intuition is wrong) to help the reader understand what this is all about, before diving into highly technical details. If my assumption about the role of the dwell time is correct, it would naturally make sense to add a sentence about systems that are tailored to display special scattering anomalies (exceptional points, self-oscillation, (coherent) perfect absorption) because these special settings boost the sensitivity even further. In addition to the previously proposed references, I can also recommend:

Chen, W., Özdemir, Ş. K., Zhao, G., Wiersig, J., & Yang, L. (2017). Exceptional points enhance sensing in an optical microcavity. Nature548(7666), 192-196.

and

 https://arxiv.org/abs/2012.10402 

To summarize, a thorough well-written discussion of context is not "beyond our scope" but essential for any good paper.

Response 2: Our work concerns the broadband complex permittivity characterization of liquids, which relies on measuring the S-parameters of the sample in the test fixture in the finite number of points (usually <1000). Due to such a sampling of the permittivity characteristic in the frequency domain, we can talk about a macroscopic liquid characterization essential for the broadband microwave dielectric spectroscopy [see Baker-Jarvis, J. et al. Measuring the Permittivity and Permeability of Lossy Materials: Solids, Liquids, Metals, Building Material, and Negative-Index Materials]. Then, one can represent such characteristics with, e.g. Debye model for engineering applications mentioned in line 14.

The terms and phenomena, suggested by the reviewer as being of importance, belong to the field of permittivity sensing, which is mainly focused on explaining anomalous permittivity changes with phenomena ongoing in a microscopic scale. Since the changes are observed in relatively small bandwidths, we agree with the reviewer that one needs to employ resonators and relevant measurement techniques, to this end. In contrast to this, our technique regards very broadband characterization of the liquid permittivity and is based on quite different instrumentation. From this aspect, our approach can be classified as macroscopic and thus we do not pretend to be experts in the permittivity sensing. Therefore, this topic is beyond our paper.

Point 3: In addition, there are some language issues that need to be sorted out before publication. I cannot list all problems here, but to give a representative example, here is one:

"To get rid of the meniscus effect authors of [20–22] applied a cell ..."

should read

"To get rid of the meniscus effect, the authors of [20–22] applied a cell ..."

Response 3: Thank you for this remark. We corrected this and also other issues found (marked in the “paper20_differences.pdf” file).

Thank you again for your review.